# Lipid Rafts and Plant Gravisensitivity

**DOI:** 10.3390/life12111809

**Published:** 2022-11-07

**Authors:** Elizabeth L. Kordyum, Olga A. Artemenko, Karl H. Hasenstein

**Affiliations:** 1Department of Cell Biology and Anatomy, Institute of Botany NASU, Tereschenkivska Str. 2, 01601 Kyiv, Ukraine; 2Biology Department, University of Louisiana at Lafayette, Lafayette, LA 70504-3602, USA

**Keywords:** cell membrane, clinorotation, microdomains, sterols

## Abstract

The necessity to include plants as a component of a Bioregenerative Life Support System leads to investigations to optimize plant growth facilities as well as a better understanding of the plant cell membrane and its numerous activities in the signaling, transport, and sensing of gravity, drought, and other stressors. The cell membrane participates in numerous processes, including endo- and exocytosis and cell division, and is involved in the response to external stimuli. Variable but stabilized microdomains form in membranes that include specific lipids and proteins that became known as (detergent-resistant) membrane microdomains, or lipid rafts with various subclassifications. The composition, especially the sterol-dependent recruitment of specific proteins affects endo- and exo-membrane domains as well as plasmodesmata. The enhanced saturated fatty acid content in lipid rafts after clinorotation suggests increased rigidity and reduced membrane permeability as a primary response to abiotic and mechanical stress. These results can also be obtained with lipid-sensitive stains. The linkage of the CM to the cytoskeleton via rafts is part of the complex interactions between lipid microdomains, mechanosensitive ion channels, and the organization of the cytoskeleton. These intricately linked structures and functions provide multiple future research directions to elucidate the role of lipid rafts in physiological processes.

## 1. Introduction

Our understanding of microgravity’s effects on living systems, including plants, improves as the result of better methods and novel technologies but also by using specific (plant) species. Examining increasingly complex interactions between specific plant responses and environmental conditions elucidates the complex effects of gravity on plants and their response to weightlessness. The realization that plants must be a component of Bioregenerative Life Support Systems (BLSS) focused on plant productivity and the identification of the most suitable species for space cultivation [1,2]. Plant functionality during long-term space missions includes the responses to elevated CO_2_, the hydrolysis of water, oxygen generation, possibly hydrogen as generic fuel, and the recycling of water. However, the provision of edible biomass is certainly the most sought-after benefit of plant growth in space [3,4,5,6]. Human explorations of the Moon or Mars are impossible without BLSSs as the reduction in resources (water, food, and oxygen), achievable with BLSSs, is essential because of energy requirements for food transport, stowage, as well as the additional mass of packaging, which makes in situ food production and BLSSs essential for long-term space exploration. Additional criteria for in situ food production are logistical and space- and energy-saving measures. The degradation of food during long-term storage and the required volume can only be solved by means of preparing fresh food.

The focus on BLSSs is different from fundamental biological questions of the effect of weightlessness on plant physiology. From the earliest to current space experiments, the effects of weightlessness were the focus of much basic research that included considerations of cellular structures, such as the organization of cells, responses of the cytoskeleton and metabolism, stress responses, and differential gene expressions between space experiments and ground controls [7,8,9,10,11,12,13,14,15,16].

Despite the fact that biological membranes, notably the cell membrane (CM), are most sensitive to altered gravity [17], there are few published data on the effect of microgravity on their physicochemical properties. The CM is one of the most dynamic supra-molecular structures in a cell, as it is the link between the cytoplasm and the extracellular matrix. It participates in numerous basic cell processes such as the transport of metabolites and ions, cell signaling, endocytosis, cell division and differentiation, and defense from pathogens [18,19,20,21,22,23]. The ability of plants to remodel membrane lipids and protein composition plays a crucial role in the adaptation to stress and environmental changes, including altering temperature, drought, salinity, and heavy metals [24].

The involvement of CM in gravisensing has been demonstrated initially by the formation of complex and diverse pattern folds in the green alga Chlorella vulgaris in space flight in comparison with ground control [25]. The appearance of CM folds of a complicated configuration during space flight was suggested to associate with enhanced metabolism and endo- and exocytosis. Hanke [26] used planar lipid bilayers with alamethicin-induced pores to demonstrate the dependence of conductivity relative to the gravity vector. The influence of gravity has been confirmed in genuine biological membranes of *E. coli* [27]. The gravity dependence of the viscosity of artificial and cell membranes was demonstrated during parabolic flights by means of fluorescence polarization microscopy [28]. Similarly, changes in the composition of phospholipids, fatty acids, and sterols in the CM of clinorotated pea seedlings have been reported [29,30,31] and prompted the hypothesis of gravitational decompensation, which postulates that the surface tension of the membrane under microgravity conditions can induce changes that are enhanced by the heterogeneity of the membrane. Thus, alteration in the physicochemical properties of the membrane leads to changes in the permeability and ion transport activity of membrane-bound channels, followed by adjustments of metabolism and eventually physiological responses [32].

The presence of functional microdomains with specific localization and lipids and protein composition in the CM became known as “membrane microdomains” or “detergent-resistant membranes” (DRMs). However, the most common name is “lipid rafts” with the distinction of rafts, clustered rafts, DRMPs, and Caveolae [33]. The formation of ordered domains was linked to the behavior of DRMs in liposomes and cells [34]. Upon stimulation, rafts may cluster to form larger structures, i.e., temporally and spatially organize functionally different protein complexes, that improve the synchronization and specificity of cellular responses [33]. The functional significance of lipid rafts, their dynamic assembly, and the enrichment of cholesterol, sphingolipids, and saturated fatty acids affect membrane trafficking and signaling. Thus, lipid rafts take part in many vitally important cell processes [20,23,35,36,37,38,39,40,41,42]. The concept of cholesterol clustering helps explain its effect on membrane fluidity via the formation of ordered domains and changed membrane deformation by altered protein interactions with the cortical cytoskeleton, and the formation and stabilization of lipid–protein assemblies [43]. The main function of sphingolipids is their involvement in signal transmission from the outer to the inner membrane surface. This function depends on various lipophilic (ceramide) or hydrophilic (carbohydrate) attachments, which determine the extent of the interaction of the lipid bilayer with the polar environment. Sphingolipids are localized on the outer membrane surface, and the specificity is determined by the attached carbohydrate moiety [44,45,46]. Lipid rafts explain the biochemical processes of the cell membrane under normal and stress conditions that cannot be explained otherwise. The increased cholesterol and saturated fatty acid content of the raft fraction in clinorotated pea seedling root membranes indicated raft sensitivity to clinorotation [47].

The aim of this review is to assess available data on plant lipid rafts as possible triggers of changes in cell activities in response to changes in mechanical stimulation and unloading.

## 2. Lipid Rafts in Plants

Basic information on the structure, composition, and functions of lipid rafts in the CM was obtained in studies of the membranes of animal and yeast cells. Later, data on the presence of microdomains enriched in sphingolipids, cholesterol, and nonionic detergent insoluble in domains, similar to the lipid rafts of mammalian cells, were reported in plants [40,41,48,49,50,51,52,53,54,55,56,57].

Lipid rafts were documented in the CM isolated from *Nicotiana tabacum* leaves and BY2 cell cultures [38,39,48,51,58]. Microdomains have sizes in the range of 100–400 nm and often contain aggregated protein complexes. Six proteins of 22, 28, 35, 60, 67, and 94 kDa anchored to glycosylphosphatidylinositol (GPI) were detected [48]. Based on one- and two-dimensional gel electrophoresis, mass spectrometry, and immunoblotting, DRMs were shown to be highly enriched in glucosylceramide and in a mixture of phytosterols—stigmasterol, sitosterol, 24-methylcholesterol, and cholesterol—compared with the CM from which they were isolated [38,39]. Phospho- and glycoglycerolipids were detected in lipid rafts in small quantities. The role of plant rafts, defined as the sterol-dependent ordered assemblies of specific lipids and proteins in plant CM as a signaling entity is based on their ability to recruit specific membrane proteins that elicit specific signaling and responses to biotic and abiotic stress, cellular trafficking, and cell wall metabolism [39,51,59]. The enrichment of microdomains with polyphosphoinositides and saturated fatty acids also suggests lipid raft involvement in cell signaling [58].

Similarly, DRMs isolated from the CM of *Arabidopsis thaliana* callus, *A. thaliana*, and *Allium porrum* leaves were enriched in sterols and sphingolipids 4–5-fold in comparison with the entire CM, sterylglucosides, and glucosylceramides and were depleted in glycerophospholipids [49,53]. Microdomains from *A. thaliana* cotyledons were enriched with specific signaling components, particularly with seven receptor kinases with leucine-rich repeats, 10 other kinases, the β subunit of heterotrimeric G-proteins, and five other GTP-binding proteins [60]. Sterol-dependent proteins associated with rafts comprise part of the membrane-bound ABA signaling system in *A. thaliana* leaves, in particular, phosphatase ABI1 (a negative regulator of ABA signaling) and calcium-dependent protein kinase 21 [61]. The glycoprotein At-FLA 4 (fasciclin-like arabinogalactan protein 4) positively regulates the biosynthesis of cell wall components and normal root growth through an ABA-dependent signaling pathway. Fasciclins are normally associated with the outer surface of lipid rafts in the CM via glucosylphosphatidylinositol [62].

In the CM of *Medicago trunculata*, lipid rafts were structurally identified as small (40–120 nm), heterogeneous, highly dynamic domains enriched in sterols and sphingolipids. Small rafts can sometimes form larger units through protein–protein and protein–lipid interactions. A specific set of proteins, including the presence of a redox system around cytochrome B561, was found in rafts isolated from the root CM. Likely, the redox system is involved in the symbiotic interaction between legumes and symbionts [52,63].

The involvement of Bax inhibitor (BI-1), which is a suppressor of apoptosis, in sphingolipids metabolism suggests a regulatory mechanism of rafts for cell death. BI-1 overexpression inhibits stress-induced cell death by decreasing the abundance of a set of key proteins. The influence of BI-1 on cell death-associated components in sphingolipid-enriched microdomains of the *Oryza sativa* CM suggests a novel, biological implication of plant lipid rafts in stress-induced cell death [64]. Higher proportions of sterols, sphingolipids, and saturated phospholipids were also detected in the microdomains of *Avena sativa* and *Secale cereale* in comparison with those in the CM [65]. Numerous proteins were identified as DRM proteins in these species. The physicochemical properties of the proteins and the unique distribution of proteins in the DRMs may control the functions of the domains in the CM for various physiological processes [66].

The involvement of lipid rafts and apoplastic transport through plasmodesmata is likely controlled by callose, which is synthesized by callose synthases and degraded by β-1,3 glucanases. The proximity of callose-modifying glycosylphosphatidylinositol proteins at plasmodesmata may depend on and be stabilized by sterols [67]. Thus, lipid rafts may affect callose accumulation in plasmodesmata and participate in the maintenance of plasmodesmal ultrastructure, callose deposition, and signaling. Therefore, lipid rafts may affect the function of plasmodesmata [68,69].

CM lipids of the cell suspension cultures of *Populus trichocarpa* are enriched in proteins of functional categories such as transport, signal transduction, responses to abiotic and biotic stress, and the biosynthesis of callose. It is especially interesting and important to note that microdomains are enriched with ABC transporters, aquaporins, sugar, metal, organic solute transporters, and ATPases, as well as phospholipase D and diacylglycerol kinases. These enzymes are involved in the biosynthesis of phosphatidic acid [70], which regulates and amplifies many cellular signaling pathways and functions, as well as in membrane arrangement [71].

The role of lipid rafts in the protective mechanisms under various stresses has been shown for cold stress in *A. thaliana* [72]. P-type H^+^ATPases, aquaporins, and endocytosis-related proteins increased and, conversely, tubulins, actins, and V-type H^+^-ATPase subunits decreased in DRMs during cold acclimation. The functional categorization of cold-responsive proteins in DRMs supports the notion that plant CM microdomains affect membrane transport and cytoskeleton interactions in *Avena sativa* and *Secale cereale* [65], drought, and salinity in *Brassica oleracea* L. var italica [73] and iron deficiency in *Beta vulgaris* [74]. A close relationship between *Gossypium hirsutum* cv. Jimian 14 fiber cell development and cell membrane lipid organization and lipid raft activity were illustrated by di-4-ANEPPDHQ-labeled fibers [75]. The raft-specific lipids enriched with cerebrosides and sterols in the halophytes *Artemisia santonica* and *Salicornia perennans* indicate the participation of rafts in salt resistance [76]. In addition, a study of the lipid raft fraction from pea seedling root CM after clinorotation showed significantly increased cholesterol and saturated fatty acids [47], which will be illustrated in more detail below.

## 3. Lipid Rafts under Clinorotation

Lipid rafts from the root CM of pea seedlings grown in stationary conditions and under slow horizontal clinorotation (2 rpm) have the appearance of thin tapes of 80–100 nm in length and 6–13 nm in width (Figure 1); they were similar to those in other plant species in structure and size and also enriched with cholesterol and saturated fatty acids.

Under stationary conditions, saturated fatty acids prevailed in the lipid raft fraction (64.5%) vs. 43.5% in the static CM. In clinorotated fractions, the percentages of the monoenoic and polyenoic fatty acids were 6.5% and 29%, respectively, and the content of saturated fatty acids increased (66.7%), monoenoic fatty acids decreased (5.2%), while the content of polyenoic fatty acids remained unchanged (28.2%). The content of cholesterol increased about seven times under clinorotation compared to the control, but the content of other sterols remained unchanged (Figure 2) [47].

The level of membrane cholesterol is important for the structure and stability of dense microdomains. A significant increase in the cholesterol content is a marker of the increased rigidity of lipid rafts that reduces membrane permeability. The regulatory role of sterols in membrane structure and fluidity and the assembly of specific proteins will affect cell responses to abiotic stress and microgravity [22,54,77]. The concept of cholesterol clustering helps explain functions such as membrane fluidity via lipid ordering and membrane elasticity by protein interactions with the cortical cytoskeleton and the formation and stabilization of rafts themselves [43]. Interestingly, the equality of the CM fatty acid saturation index (the ratio between unsaturated and saturated fatty acids) for static and clinorotated samples was most likely based on new specific fatty acids that maintained CM fluidity (microviscosity) [31,78].

### 3.1. Filipin Staining

Fluorescent marker of sterols—filipin (a complex of four pentanes [79] from the actinomycete *Streptomyces filipinensis*)—is an antibiotic that is strongly fluorescent with a maximum emission at 482 nm (λex = 340 nm) upon binding to sterol-containing areas of cell membranes. Filipin fluorescence is the established tool for sterol visualization in biological membranes [50,80,81,82,83,84,85]. After filipin staining, the CM of root tip cells of 3-day-old pea seedlings acquired the characteristic blue color (Figure 3A) [86]. In 6-day-old seedlings, the CM was marked by filipin with a dashed thinner blue line (Figure 3C) and the intensity of the filipin fluorescence diminished, possibly because the cholesterol content decreased, as changes in the cholesterol labeling intensity depend on the developmental stage [87].

Under clinorotation, the filipin labeling of the CM in 3-day-old pea seedlings appeared interrupted, which may either indicate the uneven distribution of cholesterol (Figure 3B) or the uneven binding of the dye. After 6 days of clinorotation, the filipin-stained areas were significantly larger than in the controls, clusters formed, and fluorescence increased (Figure 3D). These data suggest an increase in the cholesterol content of membrane microdomains and the clustering of microdomains under clinorotation.

### 3.2. Laurdan Staining

Laurdan (or 6-propionyl-2-(dimethylamino)naphthalene), is a fluorescent, lipophilic probe and is widely used to study the structure of biological membranes [88] in addition to styryl dyes [89]. Its emission spectra depend on the parameters of the surrounding medium, such as polarity, hydration, and viscosity. Rafts form a liquid-ordered (Lo) phase on the membrane bilayer surface, which is surrounded by a liquid-disordered (Ld) phase [90,91]. Because the Lo phase is characterized by a higher level of lipid packing compared to the Ld phase, it is also less hydrated and more viscous [92,93]. The difference in the CM water content can be estimated [94,95].

The dynamics of the Lo and Ld phases of the CM under clinorotation were examined using laurdan as an imaging probe [86,96], developed for the observation of live plant cells. No significant differences were observed in the CM of the root tip cells of the control or the clinorotated seedlings in areas with different water contents.

The expansion of the Lo areas (Figure 4) indicates higher rigidity, possibly the result of denser domains (rafts) under clinorotation.

The significantly increased cholesterol content in lipid rafts and their increased density under clinorotation suggest changes in the associated proteins. Altered CM permeability is likely to affect cell metabolism as a response to mechanostimulation. These data suggest that clinorotation does not serve as a substitute for microgravity but that plants respond to enhanced mechanostimulation. This notion supports the observed effect of hypergravity on membrane lipid composition and the conclusion that gravity resistance depends on lipid rafts [97] as well as observations that clinorotation decreases but weightlessness increases amyloplast size [98].

## 4. How Do Lipid Raft Studies Advance the Understanding of the Gravisensitivity of Plant Cells?

Plant lipid rafts have been defined as the sterol-dependent ordered assemblies of specific lipids and proteins in the plant CM. The dependence of membrane function on their lipid composition and associated proteins can affect the activity of integral membrane proteins, including ion channels [99]. According to current concepts regarding the heterogeneity of the lipid bilayer, rafts link the CM with the cytoskeleton and affect cell polarization and thus signal transmission from the membrane surface to intracellular structures, in particular, the reorganization and dynamics of the actin cytoskeleton. Proteomic analysis revealed the inclusion of various actin-binding proteins as well as the actin and α- and β-chains of tubulin in rafts [51,100,101]. The influence of cholesterol on the actin cytoskeleton depends on the initial state of the microfilament network. Cholesterol-dependent rearrangements of the cytoskeleton are determined by the balance of globular and fibrillar actin in the cell, which demonstrates a new role of membrane cholesterol in cellular mechanotransduction and organization. The suggestion that the cytoskeleton may be a gravity sensor has been supported by its modifications in microgravity and clinorotation [102,103,104], especially in non-specialized cells [104]. For example, an increase in cortical actin microfilaments and the appearance of short and disoriented “scattered” microfilaments under clinorotation were described in the root and hypocotyl elongating cells of *A. thaliana* [104,105,106]. Possibly, clinorotation is assumed to enhance the mutual dependence between the actin and tubulin elements of the cytoskeleton and part of the cellular stress response [105,106].

Cortical microtubules, which regulate the movement of the cellulose-synthesizing complexes and control cellulose deposition in the cell wall [107,108,109] changed their orientation, were more randomly distributed and often shorter in *A. thaliana* hypocotyls in space flight and *Beta vulgaris* roots under clinorotation than in the controls [104,105,106,107,108,109,110,111]. Reduced tubulin gene transcription might have caused the suppression of MT reorientation. It is interesting to note that changes in the microtubule orientation and length occurred in the elongating cells of hypocotyls and the root distal elongation zone (DEZ), distinguishing specific physiological properties and functions [103,112,113,114]. Cytoskeletal elements regulate both the growth polarity and the maintenance of cellular growth [113,115]. Cell growth in the DEZ involves cytoplasm expansion (non-directional growth), which requires the dynamic formation of MT, which makes especially cortical MT more sensitive to changes related to microgravity. These microtubules function via associated proteins and regulatory kinases and phosphatases [116].

The sensitivity of the plant cell wall structure, composition, and function is well-established. [7,13,117]. The general phenomenon of cell wall thinning and softening in microgravity and clinorotation could be related to qualitative and quantitative changes in the poly- and monosaccharide composition based on the increased activity of cellulosopectolytic enzymes, for example, endo-1,4-B-glucanase, exo-1,4-B-glucanase, polygalacturonase, and pectin esterase. Such changes were demonstrated in the protonemata of *Funaria hygrometrica* grown under long-term slow clinorotation and in wheat plants grown in space [117]. Galacturonic acid reduced the methoxyl groups of pectins, which release calcium ions because of enhanced cellulose and pectin hydrolysis. Decreasing levels of cellulose and polysaccharides per unit of length have also been reported in the coleoptiles and roots of the *Oryza sativa* and *A. thaliana* hypocotyls in spaceflight compared with ground controls [13,118,119]. The degradation of 1,3:1,4-β-glucans reduces these polysaccharides in growing rice shoots and affects the mechanical properties of cell walls in microgravity. However, microgravity only slightly affected cell wall biopolymer synthesis and the deposition of cellulose microfibrils [120].

Modifications of the intracellular calcium concentration and localization were shown in microgravity and under clinorotation similar to the fluorescent calcium indicators chlorotetracycline and indo-1. Changes in the relative calcium ion content have been demonstrated in the cells of the green algae *Chlorella vulgaris*, the protonemata of *Funaria hydrometrica*, the statocytes of *Pisum sativum*, *Melilotus album*, and *Glycine max*, the microcalluses of *Daucus carota* and *Brassica napus*, as well as in the cytosol of the root hairs in *Lepidium sativum* and *Beta vulgaris*, and in the callus cell cultures of *A. thaliana* [121,122,123,124]. Most experiments showed that [calcium] in cells increased under microgravity and clinorotation. Calcium is a well-established second messenger and plays a crucial role in signal transduction in all eukaryotic cells. Its messenger functions involve transient changes in the cytosolic ion concentration in response to a variety of external and internal stimuli, including light, hormones, temperature, anoxia, salinity, and gravity [125,126,127]. The role of calcium ions as secondary messengers depends on short-term alterations in cytosolic concentration, which are mediated by the actions of Ca^2+^-permeable ion channels, the efflux by Ca^2+^-ATPases, and Ca^2+^/H+ exchangers [128,129,130]. Calcium ions enter the cytosol from the apoplast through selective channels in the CM and/or a release from intracellular sources, such as an endoplasmic reticulum (ER) or the vacuole, which are typically activated by inositol-1,4,5-triphosphate [131]. An increase in the intracellular calcium activity in microgravity was assumed to occur due to the activation of mechanosensitive calcium channels, which was demonstrated by gadolinium, which blocks such channels [132,133], as well as a possible decrease in Ca^2+^-ATPase activity associated with lipid rafts [134]. The complex interactions between the roles of cholesterol and lipid microdomains in regulating mechanosensitive ion channels and the organization of the cytoskeleton provide multiple future research directions.

These prospects for further studies of membrane lipid microdomains include their participation in cell signaling and cell membrane activities as part of experiments that seek to understand the effect of gravity and mechanostimulation on basic cellular processes in biological systems. There is an additional temporal element in this notion as the lipid profile of membranes changes depending on the type of tissue, cell, and organelle [45,67,135]. The quantitative differences in the content of lipids and fatty acids in the CM of epicotyls and roots are most likely associated with the structure, growth, and specific functions of specific cells [136]. In contrast to the root proper, the root cap contains statocytes, which are specialized cells for the perception of gravitational and mechanical stimuli including clinorotation. Therefore, the CM state of these tissues exhibits a specific composition [21,67]. The heterogeneity of lipid rafts is linked to the specificity of the transcriptome [137] and proteome interactions [138]. The successful implementation of Bioregenerative Life Support Systems, therefore, benefits from a better understanding of cell membrane functions, especially lipid rafts, and their improvement of nutrient acquisition [139].

## 5. Conclusions

The data and inferences described so far clearly indicate that the functional domains of membranes are critically important for signal integration and responses. We hope that this review will prompt future research on the effect of lipid rafts and membrane functions on general and gravitational physiology.

## Figures and Tables

**Figure 1 life-12-01809-f001:**
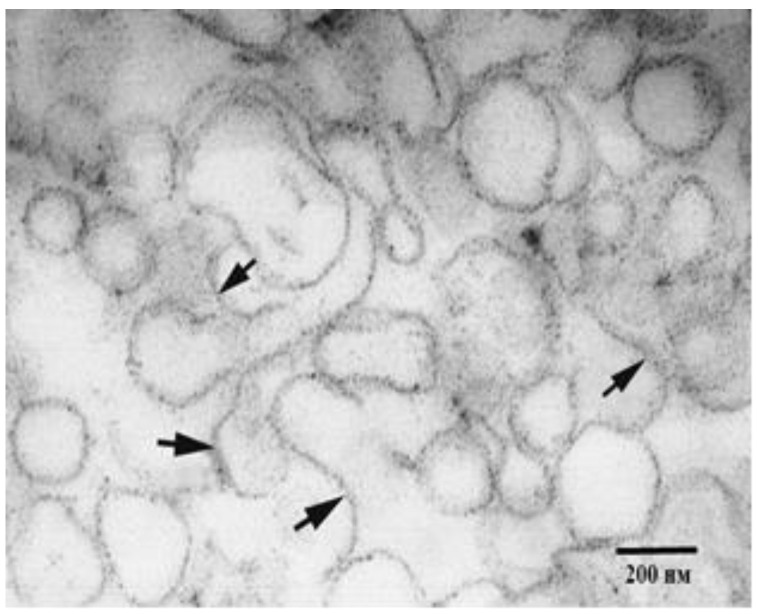
Raft fraction isolated the root CM of 6-day-old pea seedlings (transmission electron microscopy). Arrows point to rafts [47].

**Figure 2 life-12-01809-f002:**
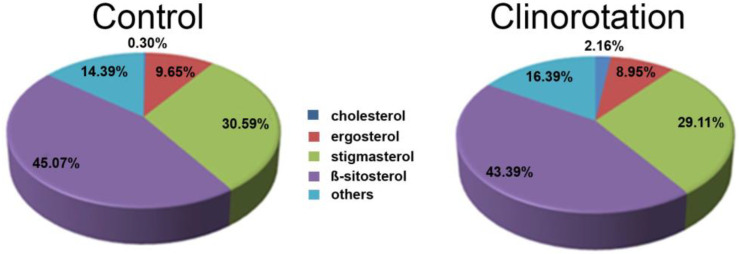
Data on the sterol content in the raft fractions isolated from the root CM of six-day-old pea seedlings grown statically (control) or after clinorotation [47].

**Figure 3 life-12-01809-f003:**
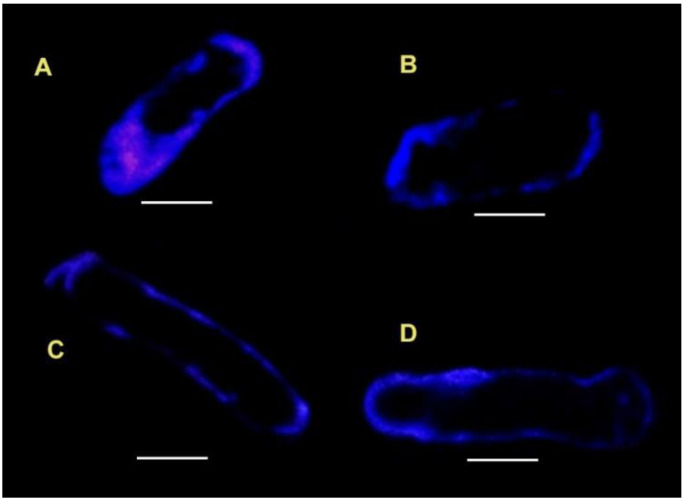
Images of live, elongating root cells of 3-day-old (**A**,**B**) and 6-day-old (**C**,**D**) pea seedlings stained with filipin staining: (**A**,**C**)—control; (**B**,**D**)—clinorotated. Scale bar 20 μm [86].

**Figure 4 life-12-01809-f004:**
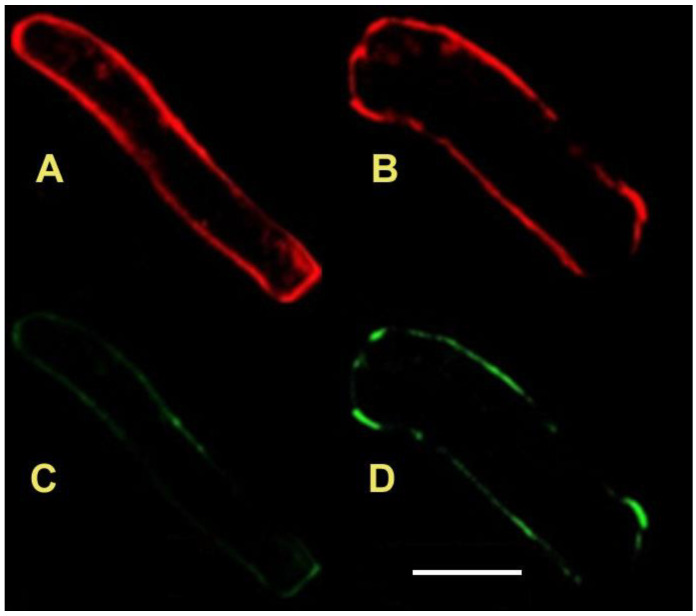
Images of live 6-day-old pea root cells stained with laurdan without (**A**,**C**) and after clinorotation (**B**,**D**). Red fluorescence (620–650 nm) indicates higher water content, representing disordered (Ld) state. Blue-shifted emission (480–500 nm, green) indicates lower water content and ordered (Lo) state. Scale bar 50 μm [86].

## Data Availability

Not applicable.

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
