# Peer review of "Lipid Rafts and Plant Gravisensitivity"

_life, 2022, doi:10.3390/life12111809_

Round 1

Reviewer 1 Report

The structural continuum or physiological continuity of cytoskeleton-plasma membrane-cell wall plays a principal role in plant response to gravity signal, and space experiments related to this topic have been conducted. Out of various constituents of the plasma membrane, microdomains called lipid rafts are believed to be responsible for important cellular processes such as signaling, trafficking, and structure formation. However, information on the role of lipid rafts in plant gravity responses is unorganized. The present ms is a nice review for this topic and has a fundamental value to be published in Special Issue 'Plants and Microgravity' in Life. I would suggest only two points to improve the ms further.

1. If lipid rafts are responsible for gravity signal integration and responses, it is expected that their formation and activities are more or less suppressed under microgravity conditions. However, the changes caused by clinorotation appear to be opposite to this expectation. Does it suggest the limitation of clinostat as a microgravity simulator?

2. The importance of plant production and BLSS in space explorations is mentioned in the first part of Introduction, but there is no description after that. It is advisable to discuss the possible contribution of lipid raft research to plant cultivation in space.

Author Response

Wondered if a clinostat is a limited microgravity simulator.

The reviewer is correct. Clinorotation is not a substitute for weightlessness but adds mechanostimulation to the system.

We included the following statement to the text (L 259):

These data suggest that clinorotation does not serve as a substitute for microgravity but that plants respond to enhanced mechanostimulation. This notion supports the observed effect of hypergravity on membrane lipid composition and the conclusion that gravity resistance depends on lipid rafts 97 as well as observations that clinorotation decreases but weightlessness increases amyloplast size 98.

Stated that other than the introduction no reference is made to Biological life support systems.

We added a concluding remark to the text (L 346): The successful implementation of Bioregenerative Life Support Systems therefore benefits from a better understanding of cell membrane functions especially lipid rafts, and their improvement of nutrient acquisition 139.

Reviewer 2 Report

The authors summarize the latest findings on lipid rafts, and their function in plant responses under simulated microgravity conditions using clinostat or true microgravity conditions in space. Long-term manned space exploration requires the cultivation of plants under microgravity conditions in space, and in recent years, more attention has been paid to the study of the graviresponses of plants than ever before. Therefore, while this reviewer considers this review is informative and useful, this reviewer has the following several comments.

Which of the direction of gravity or the magnitude of gravity affects the dynamics of the lipid rafts? Clinostats minimizes the effect associated with a unilateral gravitational stimulus, but does not actually remove the mechanical load of the gravitational force. In the microgravity conditions in space, on the other hand, both the direction and magnitude of gravity are removed. These facts suggest that the direction of gravity may affect the dynamics of lipid rafts. However, Koizumi et al. (2007) showed that membrane lipid composition is changed by hypergravity in azuki bean epicotyls (http://dx.doi.org/10.1016/j.asr.2007.02.040). Would it be possible to add the authors' interpretation in this regard to the revised manuscript?

There is an incorrect way to describe the scientific name. For example, Medicago on line 136 must be shown in italics. Also, L. var italica on line 174 should be shown in regular forms.

Line 148, phospholipids, is underlined.

In Figure 2, decimal points are indicated by commas.

Check how to cite references in the text and how to give section titles (numbering).

Author Response

The reviewer pointed out:

1) that hypergravity  leads to a change in lipid composition. 

This remark relates to reviewer 1’s comment and we have included a reference to the paper in the amended text (L 259 ff):

These data suggest that clinorotation does not serve as a substitute for microgravity but that plants respond to enhanced mechanostimulation. This notion supports the observed effect of hypergravity on membrane lipid composition and the conclusion that gravity resistance depends on lipid rafts 97 as well as observations that clinorotation decreases but weightlessness increases amyloplast size 98.

2) showed incorrect use of italic font and underlining. 

we italicized In the CM of Medicago trunculata in L136 and removed it from var. italica (L 174)

We removed the underlined (hyperlink) from phospholipids (L 148)

We also added numerical specifiers to section titles and checked cited references.